# Four-dimensional surface motions of the Slumgullion landslide and quantification of hydrometeorological forcing

Xie Hu [1,2 ✉], Roland Bürgmann [1,2], William H. Schulz [3] & Eric J. Fielding [4]

Landslides modify the natural landscape and cause fatalities and property damage worldwide. Quantifying landslide dynamics is challenging due to the stochastic nature of the environment. With its large area of ~1 km$^2$ and perennial motions at ~10–20 mm per day, the Slumgullion landslide in Colorado, USA, represents an ideal natural laboratory to better understand landslide behavior. Here, we use hybrid remote sensing data and methods to recover the four-dimensional surface motions during 2011–2018. We refine the boundaries of an area of ~0.35 km$^2$ below the crest of the prehistoric landslide. We construct a mechanical framework to quantify the rheology, subsurface channel geometry, mass flow rate, and spatiotemporally dependent pore-water pressure feedback through a joint analysis of displacement and hydrometeorological measurements from ground, air and space. Our study demonstrates the importance of remotely characterizing often inaccessible, dangerous slopes to better understand landslides and other quasi-static mass fluxes in natural and industrial environments, which will ultimately help reduce associated hazards.

[1] Berkeley Seismological Laboratory, University of California, Berkeley, CA, USA. [2] Department of Earth and Planetary Science, University of California, Berkeley, CA, USA. [3] US Geological Survey, Denver, CO, USA. [4] Jet Propulsion Laboratory, California Institute of Technology, Pasadena, CA, USA.
✉email: xiehu@berkeley.edu

Landslides denude mountains, transport sediments to rivers, lakes and oceans, and modify the Earth's surface environment and ecosystem. Landslides of all sizes and rates represent geohazards that may lead to property damage and casualties. The hazards that landslides present and their impact on Earth's surface primarily depend on their volume and the rate at which they move, as well as their responsiveness to hydroclimatic variability. However, quantifying landslide dynamics is challenging due to the stochastic nature of the environment (e.g., geology, geomorphology, and vegetation), external disturbances (e.g., fire, climate change, earthquakes, and logging), and the limited availability of observations (e.g., remote, surface and subsurface geodetic, and geophysical and hydrological measurements)[1–5]. Knowledge of landslide behavior primarily depends on isolated measurements made on and within the landslides, which are often cost prohibitive or even impossible to obtain, and their value is limited by conservative interpretations for generalizing to the entirety of dynamically complex landslides. Incomplete information of three-dimensional (3D) surface displacements has limited our ability to infer the continuous landslide depth, interpret the driving and resisting mechanisms, and develop accurate forecasts for landslides. Here, we compile a comprehensive dataset of remote sensing imagery from air and space, meteorological records, and in situ surface (extensometer) and subsurface (inclinometer) deformation measurements, allowing us to develop a systematic framework for using detailed, temporally variable 3D surface deformation data to quantify the underlying landslide kinematics and dynamics.

For centuries, the Slumgullion landslide in the San Juan Mountains of Colorado has snaked its way downhill at ~10–20 mm per day[5–16], allowing us to explore both transient and quasi steady-state mass wasting processes. The original 700-year-old failure initiated from the edge of the Cannibal Plateau, formed Lake San Cristobal, and is currently inactive (Fig. 1). About 300 years ago, a ~3900-m-long and 150- to 450-m-wide section of the landslide reactivated from the original headscarp to a new toe above Highway 149. The landslide deposits consist of hydrothermally altered Tertiary volcanic rocks.

Interferometric synthetic aperture radar (InSAR) has been widely used to measure ground motions for geohazards research[17–19], but its application at Slumgullion is challenged by high deformation gradients. In addition, the reconstruction of 3D surface displacements depends on the availability of multiple view angles and their distribution[16,20]. Here, we incorporate data from the ascending and descending tracks of Copernicus spaceborne C-band Sentinel-1 SAR (2017–2018) and four flight lines of the NASA/JPL airborne L-band Uninhabited Aerial Vehicle SAR (UAVSAR) (2011–2018; Fig. 1a) with a hybrid InSAR phase and SAR amplitude pixel offset tracking (POT) time-series analysis (Supplementary Figs. 1 and 2 and Supplementary Table 1)[21,22]. The advance in data and method integration illuminates the spatiotemporal 3D surface evolution from 1000+ individual displacement maps (Supplementary Fig. 3), two orders of magnitude more than previous SAR-based studies at Slumgullion[14–16]. Variations in recharge, mainly from snowmelt, drive multi-annual decelerations and accelerations, during which

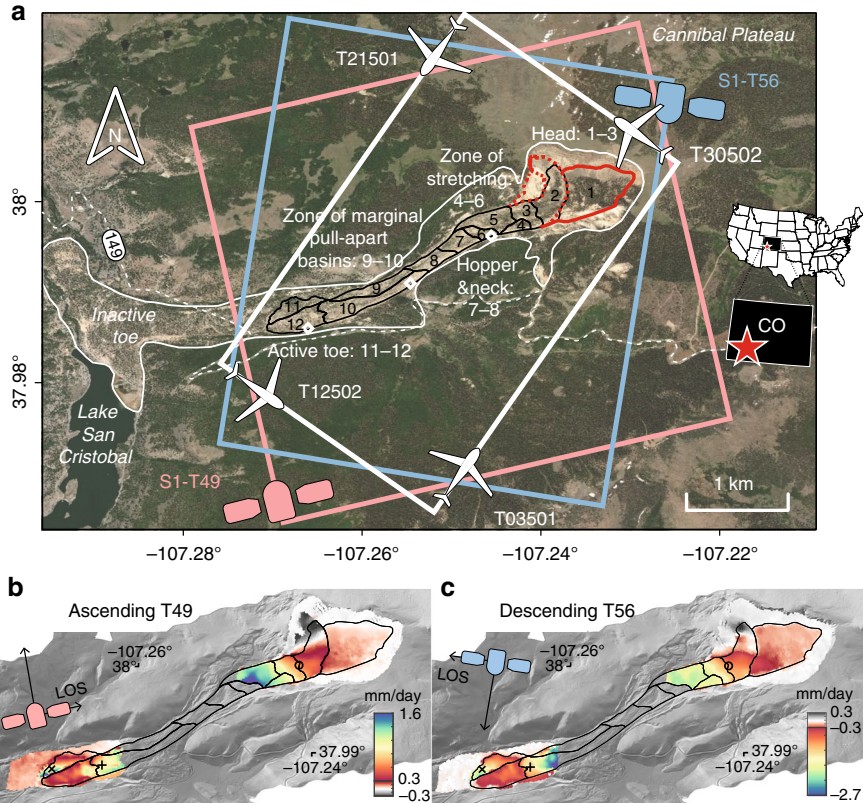

**Fig. 1 Map of the Slumgullion landslide. a** Landslide landscape. Red and blue boxes show swaths from Sentinel-1 ascending (T49) and descending (T56) tracks, respectively. White box shows UAVSAR swaths. Black and gray lines outline the active and inactive landslides. White diamonds show the locations of three extensometers; the borehole inclinometer was located near the center extensometer. Structural zones and kinematic elements are labeled[8,9,14]. Red lines show updated boundaries from this study, and dashed lines are tentative. **b, c** Sentinel-1 line-of-sight (LOS) displacements positive towards the satellite, superimposed on the shaded relief light detection and ranging (LiDAR) digital elevation model (DEM). Three symbols (o\x\+) show targets with time-series plots in Fig. 4a.

the head of the landslide is the most responsive. The power-law flow theory helps reconcile the mass movement with the sub-surface geometry which is explicitly characterized by a novel description of the landslide thickness, the steepness between the lateral and bed shear surfaces, and the tilt of the basal bed.

## Results

**Spatial displacement patterns.** The 98 scenes of Sentinel-1 SAR reveal displacement details over the more slowly deforming head and toe areas of the landslide (Fig. 1b, c). The fast-moving middle parts are not resolvable due to extreme InSAR phase gradients at the available Sentinel-1 orbital period, radar wavelength, and the amount of displacement over a short distance (see "Methods" for details). We refine the boundaries of a kinematic element in an area of ~0.35 km[2] below the crest of the prehistoric landslide (Fig. 1a), which accounts for ~1/3rd of the previously mapped mobile area[8,9,14]. We further update the structural zones[9,14]: head zone (kinematic elements #1–3) exposed by extensional fractures, zone of stretching (#4–6) characterized by broad bands of tension cracks and normal faults, hopper and neck (#7–8) resembling a funnel, zone of marginal pull-apart basins (#9–10) accompanying widening of the slide, and toe (#11–12) overriding inactive sur-faces. The current major source of debris supply appears to be on the upper flank of the head (blackish area in Fig. 1b, c with

motion to east). Here, the sediments are transported along a curved track parallel to the margin between elements 1 and 2, at a large angle from the main stream of the slide.

To systematically analyze the kinematics and mechanics of Slumgullion, we rely on 3D velocity fields that describe the steady state, slow-moving earth flow. We obtain eight velocity measure-ments from four UAVSAR flight lines during each sortie in their respective azimuth and range directions. The hybrid InSAR-POT analysis provides us a robust 3D solution over the entire active landslide area, with a total of 124 scenes. We represent the deformation in a series of 77 transverse profiles (Fig. 2a and Supplementary Fig. 4). The displacements in the steep upper head zone are highly variable with low signal-to-noise ratio (SNR). Velocity measurements become more coherent from the inter-section between the head and the zone of stretching, moving at about 2.5 mm per day. The south part of the zone of stretching moves at 7 mm per day, and elongated flank ridges extend along its southeastern lateral margin (Supplementary Fig. 4). The movement rotates westerly to the narrow hopper and neck. The velocity profiles regain a symmetric pattern with rates as high as 13 mm per day at the center. The surface topography gradually develops a bump along the central axis (Supplementary Fig. 4). The rates decrease to ≤10 mm per day in the zone of marginal pull-apart basins, and the velocity profiles appear asymmetric around the internal bends. An oversteepened northwest-facing

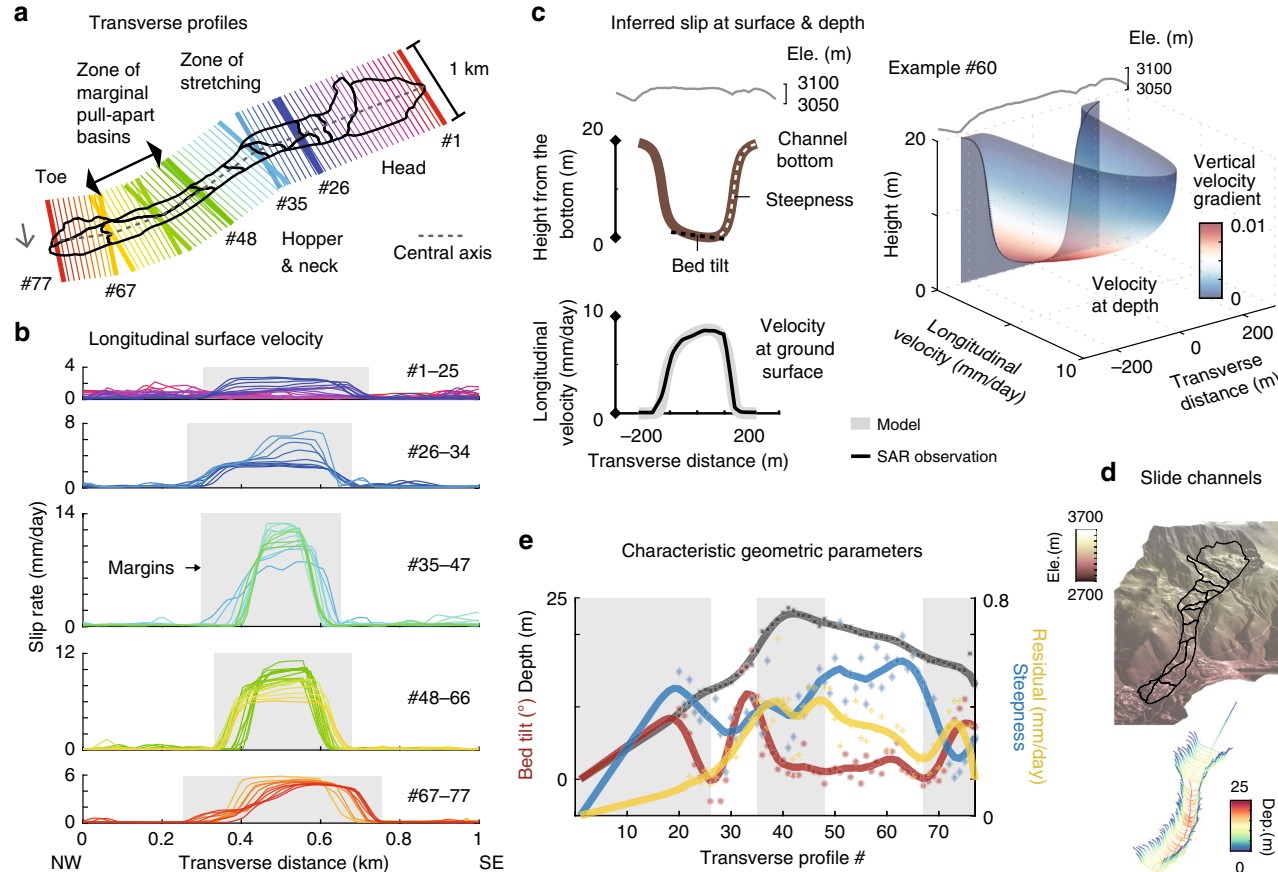

**Fig. 2 Surface and subsurface landslide kinematics. a** Transverse profiles from #1 to #77 selected every 50 m and differentiated by colors; the dashed gray line delineates the central axis. **b** Longitudinal surface velocity from hybrid InSAR-POT analysis of UAVSAR data; colored lines represent 1-km-long profiles in (**a**); the slip rate for five structural zones is plotted separately with their corresponding profile lines, and the orders of the colored lines are marked to the right; shaded sections are within the lateral margins determined by the velocity profiles. **c** Example of the inferred channel geometry and slip rates at the surface (compared with SAR results) and subsurface. Geometric steepness and bed tilt are indicated by white and black dashed lines, respectively. **d** 3D view of landslide surface and the inferred basal morphology. **e** Changing characteristic geometric parameters along the landslide. Symbols are the individual results for each profile with fitting lines in corresponding colors also used for y-axis labels.

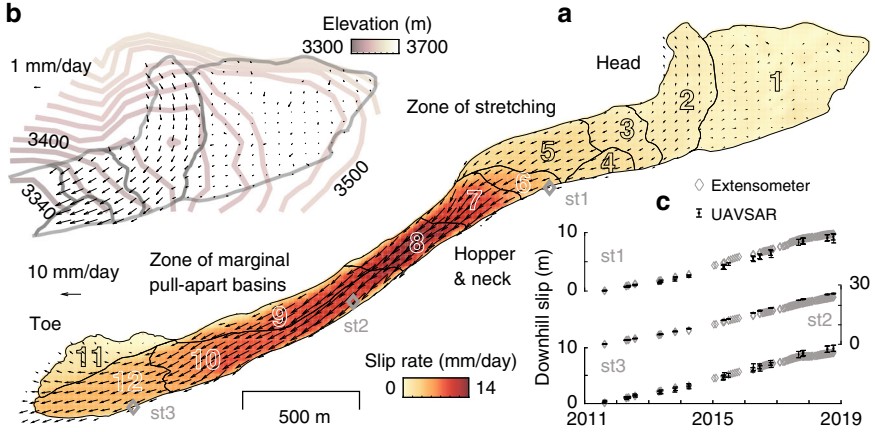

**Fig. 3 Landslide spatial dynamics. a** Horizontal slip vectors derived from 2011 to 2018 UAVSAR data with net 3D velocities in color. **b** Inset closeup view of the head also shows elevation contours. **c** Comparisons between UAVSAR time series and data from three extensometers (locations as in (**a**)). The error bars of UAVSAR represent one standard deviation of the cumulative displacements measured within a distance of ~25 m from each extensometer.

slope divides the toe, and the southern part moves faster along this internal right-lateral fault at up to 6 mm per day. The persistently advancing landslide toe results in shifted fronts with respect to those mapped in the summer of 1991[8]. Multiple pieces of evidence validate the advance of the toe over its substrate. UAVSAR-derived 2011–2018 horizontal velocities at the tip of the toe reach 4–5 mm per day, consistent with the rate determined from aerial photos taken in 1985 and 1990[9]. The mapped shifts (Fig. 3a) and the topographic back-projection (Supplementary Fig. 5) also show that the toe has advanced by ~40 m during the past two decades.

## Discussion

**Inferred landslide channels and subsurface flow.** The transverse longitudinal velocities allow us to invoke the depth-integrated law of mass conservation in order to estimate the local free-surface height from the slide channel bottom. Insignificant shear deformation down to about 10-m depth found at the borehole inclinometer indicates that the landslide materials are highly plastic and follow non-Newtonian behavior (Supplementary Fig. 6 and Supplementary Data 1). In addition, SAR and extensometer (Fig. 3c and Supplementary Data 2) measurements at lateral flanks reveal appreciable highly localized deformation that suggests a pseudo-plastic rheology at shallow depths. Therefore, we apply the power-law flow theory to characterize the upper pseudo-plug and the lower yield zone above the underlying bedrock[23] (see "Methods"). Viscoplastic flow models suggest that the longitudinal shear velocities at the surface mirror the shape of the subsurface channel[24,25]. We propose a novel geometric description of the landslide channel, which characterizes the depth, the steepness between the basal bed surface and the lateral margins, and the tilt of the basal bed across the landslide with respect to the horizontal (Fig. 2 and Supplementary Figs. 7–10).

We first focus on the emergent toe where the depth can be reliably estimated by comparing the surface topography on and off the distal slide (Supplementary Fig. 5). Compiled with the UAVSAR-measured surface velocity at the toe, we can quantify the power-law index as 0.7 and the consistency index as $1.34 \times 10^{10}$ Pa s$^n$ (Supplementary Fig. 11 and Supplementary Table 2). We then use the longitudinal surface-velocity profiles in other parts of the slide to invert for their corresponding geometric parameters (see "Methods," Fig. 2e, Supplementary Table 3, and Supplementary Movie 1). The largest depth (<~30 m) is inferred underneath the fastest-moving hopper and neck. High steepness values concentrate at the zone of marginal pull-apart basins in the

lower part of the slide. According to the inferred degree of bed tilting, the head and toe areas are more asymmetric, consistent with their irregular outlines. The bed starts from a minor NW tilt in the zone of stretching and transitions to the largest positive SE tilt at the biggest bend of the slide. Our quantification of the landslide geometry yields a total volume of $1.33 \times 10^7$ m$^3$, compared with a previous estimate of $1.95 \times 10^7$ m$^3$ (ref. [8]). We can also resolve the subsurface viscoplastic flow rate along with the channel geometry (Fig. 2c–e). High velocity gradients concentrate near the bottom of the slide and approach 0 at shallower depths.

**Hydrological forcing and time-dependent landslide deformation.** We also explore the landslide behavior controlled by the time-variant hydrological environment. Because fluid water is the essential agent that regulates the pore pressure[1,26,27], we explicitly consider the forms of precipitation and determine the daily fluid water from snowmelt and rainwater (see "Methods"). The water year 2018 (October 1, 2017–September 30, 2018) was historically dry with only 64% of 1981–2010 precipitation average. There was another dry water year in 2013 from the estimated fluid water recharge; however, this is barely discernible from raw precipitation data (Supplementary Fig. 12). From the recharge time series, we simulate pore pressures at depth as a one-dimensional diffusive process from the surface (see "Methods" and Supplementary Fig. 13).

We investigate the slide's temporal response to the estimated seasonal and multi-annual fluid water changes by comparing Sentinel-1 InSAR results of August in 2017 (wet) and 2018 (dry), for which time-series solutions are available for both tracks (Fig. 4). Between the two time periods, radar line-of-sight rates slowed down by 90% at high elevations (3450 m). The rate reduction decreases linearly from there to 70% over the upper slide (3300 m elevation), while it is only around 45% in the toe area (2950–3100 m elevation). Extensometers located on the southern flanks of elements 6, 10, and 12 show decelerations by 66%, 40%, and 49%, respectively, between the same periods (Fig. 3c). The variable rate decreases in response to the reduced water recharge imply spatiotemporal diversity of pore pressure feedback.

The UAVSAR hybrid InSAR and POT analysis captures the temporal behavior of the whole landslide during 2011–2018 at a coarser temporal sampling (Fig. 5a and Supplementary Figs. 14 and 15). The time series of downslope motions from UAVSAR and three extensometers match well (Fig. 3c). Based on the long-

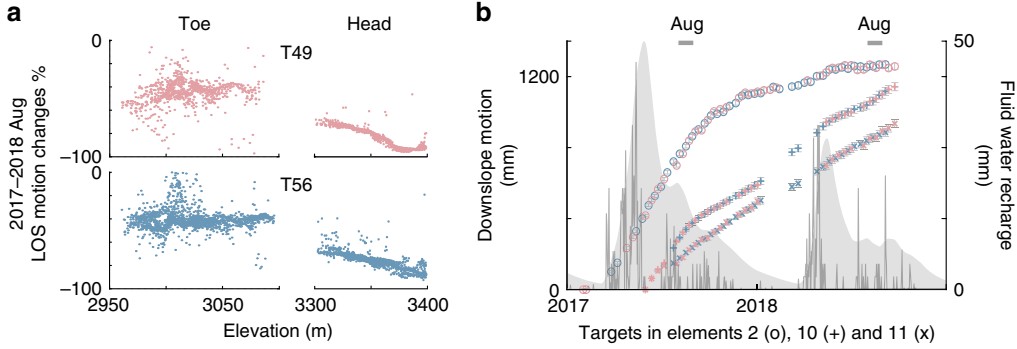

**Fig. 4 Multi-annual landslide motions from Sentinel-1 ascending (red) and descending (blue) InSAR analysis. a** The line-of-sight velocity (LOS) changes between August 2017 and 2018 over the head and toe, plotted against the elevation. **b** The downslope motion of selected targets (locations in Fig. 1). The error bars represent 1 standard deviation of the cumulative displacements measured within a distance of ~25 m from the selected target.

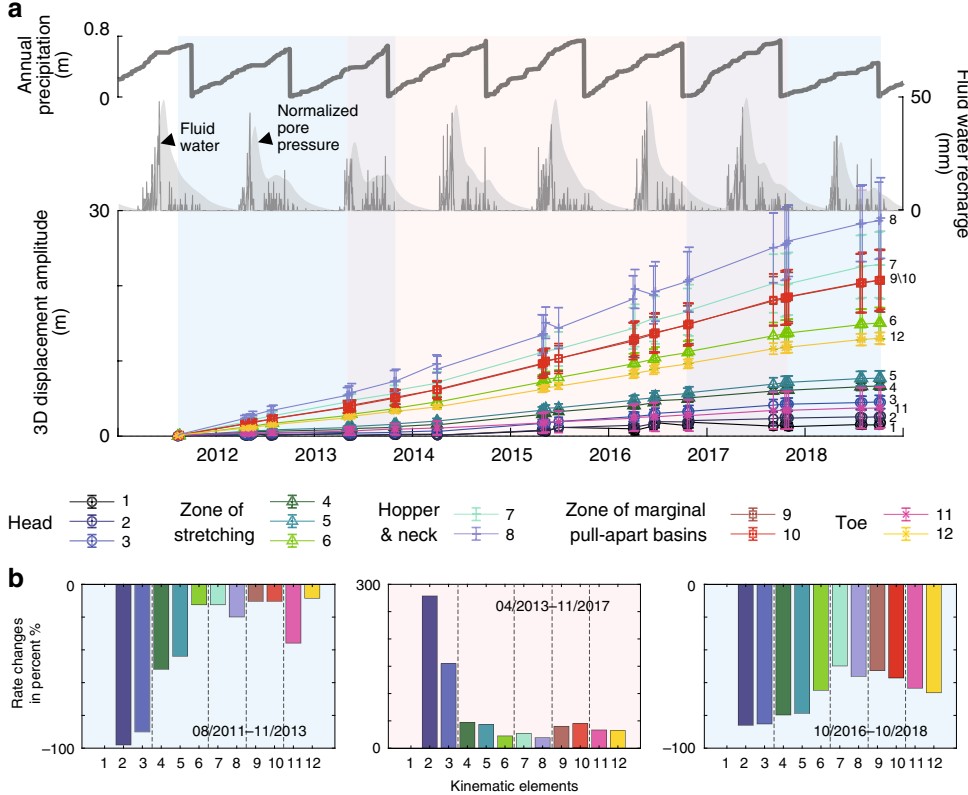

**Fig. 5 Multi-annual landslide motions from UAVSAR hybrid InSAR-POT analysis. a** The annual precipitation observations, the estimated water recharge at the surface, and the normalized pore pressure at 20-m depth are shown in gray thick, thin lines and shades, respectively. 3D displacement amplitude for each kinematic element in indicated colors and with number labels to the right (1–12). The error bars represent 1 standard deviation of the cumulative displacements for the targets in each kinematic element. Blue and red shades on the background show the time periods of observed deceleration and acceleration, respectively, consistent with the meteorological data. **b** Multi-year rate changes in three respective periods for the kinematic elements 1–12.

term changes in the fluid-water recharge history, we consider three multi-annual phases of 8/2011–11/2013, 4/2013–11/2017, and 10/2016–10/2018. We use an exponential model to quantify the multi-annual rate changes (i.e., the velocity changes with respect to initial conditions for each period, see "Methods"). Element 1 was excluded due to small SNR. Agreeing well with the hydrological processes, the inferred rate changes are negative for 2011–2013 and 2016–2018, indicative of slowing down, contrasting with the inferred speeding up during 2013–2017 (Fig. 5b and Supplementary Figs. 16 and 17). The head area consistently responds most sensitively to recharge changes during all three phases (Fig. 5b).

The distribution of UAVSAR-measured rate changes with position on the landslide is consistent with the deceleration vs. elevation/position relationship observed during 2017–2018 with Sentinel-1, as well as the extensometer data. This suggests a correlation between the landslide depth and sensitivity to hydrological forcing. This is physically consistent because the diffusive pore pressure changes more strongly and rapidly at shallow depths, while the response is increasingly damped and delayed at greater depths. For example, the onset of the pore pressure increase at 20-m depth lags behind that at 10-m depth by ~12 days according to the constrained diffusion model adjusted by the documented hydraulic diffusivity of

$7.8 \times 10^{-5}$ m$^2 \cdot$s$^{-1}$ (ref. [2]) and the inferred average pore pressure of 177 kPa (ref. [12]) (see "Methods" and Supplementary Fig. 13).

**Implications for the perennial motions of landslides.** Water recharge at Slumgullion increases twice per year, from snowmelt in late spring and rainfall in late summer and early fall, which results in a more stable nearly saturated system than landslides that experience only one significant annual recharge episode. For instance, the near-saturation condition can last for about half of the year from March at a characteristic depth of 20 m (Supplementary Fig. 13). Our results also show that over time scales of several years, Slumgullion accelerates and decelerates due to multi-year hydrological fluctuations, supporting the hypothesis that it and other landslides in the Rocky Mountains will slow in future decades due to predicted temperature increases, precipitation decreases, and a depletion of supply[13]. Other factors, such as changes in vegetation cover and possible large failures at the headscarp, could make the situation more complicated[13]. Moreover, other stabilizing mechanisms, such as shear strength that increases with shear displacement rate, shear-induced dilative strengthening, soil wetting and swelling along the lateral margins above the water table, and the forced circulation of pore fluid around asperities may help augment the resistance[27–30]. Hourly sampled subsurface strain and pore pressure data and laboratory testing may be able to identify and distinguish these contributions to the landslide strength. If such forces play a role, we can qualitatively determine that the landslide neck with large contact areas along the sides, and the zones of hopper and neck and pull-apart basins that have large irregularities in landslide depth and steepness, may provide additional stabilizing force.

## Conclusions

Unprecedented, interdisciplinary observations, methods and models combined help to advance the characterization of landslide dynamics. Remotely sensed SAR data and hybrid processing methods allow us to achieve 3D spatiotemporal surface displacements. In situ data sets such as the extensometer measurements locally validate and calibrate the SAR results from air and space; the inclinometer data provide evidence on the non-Newtonian behavior of the landslide mass and, together with the SAR/extensometer-confirmed mobility at the margins, support the application of power-law viscoplastic flow theory; the precipitation and temperature records illuminate the fluid recharge from snowmelt and rainwater; and the piezometer-measured average pore pressures help translate recharge at the surface to pore pressures at depth. Our study sheds new light on the landslide boundaries, geometry, subsurface flow, and how different structural zones respond to the hydroclimatic variability. Beyond that, our systematic chains of analysis can also be applied in full or in part to help understand other quasi-static viscoplastic flow processes associated with solid particles with an interstitial fluid, such as debris slides, volcanic lahars, and submarine slides.

## Methods

**SAR data sets.** UAVSAR is operated by JPL/NASA[21], and the L-band system has been repeatedly deployed over the Slumgullion landslide since 2011. The wavelength is $23.8 \times 10^{-2}$ m and the single-look pixel spacing along the azimuth and the range directions is 0.6 and 1.67 m, respectively. Taking advantage of the flexible trajectory of the aircraft, four independent flight lines were deployed aligned with or orthogonal to the southwesterly slip direction of Slumgullion (Fig. 1a). About 30 acquisitions are now available for each flight line from August 2011 to October 2018, representing a remarkable airborne SAR collection (Supplementary Table 1).

The C-band Sentinel-1A satellite was launched in 2014 by the European Space Agency, followed by the identical Sentinel-1B satellite launched in 2016. The wavelength is $5.547 \times 10^{-2}$ m and the single-look pixel spacing in azimuth and range directions is 13.9 and 2.3 m, respectively. Ascending track T49 and descending track T56 data sets cover Slumgullion (Fig. 1a). It was not until mid-2017 that the Sentinel-1 satellites began to collect the data over this area more

regularly with a 12-day interval. A total of 98 scenes acquired between February 2017 and September 2018 are used for this study. A larger number of interferograms are applicable for the less deforming head area compared with that of the toe (Supplementary Table 1).

**Interferometric SAR and its feasibility.** InSAR measures ground displacements from the phase difference between SAR acquisitions. One obvious challenge for InSAR at Slumgullion comes from large displacement gradients. In an ideal case with perfect coherence, the maximum detectable displacement gradient is one fringe per pixel[31,32]:

$$d_{\max} = \frac{\lambda}{2 \times \eta}, \tag{1}$$

where $\lambda$ is the radar wavelength, and $\eta$ is the pixel size. For Sentinel-1 data, the incidence angle at Slumgullion is about 46°. The minimum number of looks to scale a pixel into a square dimension on the ground is $1 \times 4$ and thus $\eta \approx 14$ m and $d_{\max} = 2 \times 10^{-3}$. The narrowest and fastest neck region of the landslide moves at ~20 mm per day and has a width of 200 m; we consider the half width (100 m) to calculate $d_x$ because the most rapidly moving part is in the center. The time interval of Sentinel-1 data is generally 12 days. This gives an actual displacement gradient of $d_{\text{act}} = 2.4 \times 10^{-3}$, which exceeds the maximum detectable displacement gradient. Thus, theoretically, Sentinel-1 data are not appropriate for Slumgullion InSAR, at least not on the fast-moving parts. Fortunately, we can still utilize the Sentinel-1 data over the less deforming head and toe areas. We use a 24-day temporal threshold, $1 \times 4$ multi-looks, and a $20 \times 20$-m filtering window over the head and toe areas (Supplementary Fig. 2).

For the UAVSAR data sets, we perform InSAR for the image pairs collected a couple of weeks apart (3–16 days). We use 12 and 3 looks in the azimuth and range directions, respectively, so that the average pixel dimension is ~7 m. The maximum detectable displacement gradient $d_{\max} = 1.7 \times 10^{-2}$. In the scenario of the longest 16-day interval, the expected displacement gradient reaches $d_{\text{act}} \approx 3.2 \times 10^{-3}$, one order of magnitude less than the maximum detectable value, suggesting that the applied number of looks works for this study.

**SAR pixel offset tracking and its feasibility.** As a complement to the InSAR method, POT can resolve large displacements in both azimuth and range directions by cross correlating the oversampled image patches. The precision can reach up to 1/20th of the pixel spacing[22,33,34]. The high spatial resolution of UAVSAR makes POT possible at Slumgullion.

Assuming that the transport zone moves as fast as 20 mm per day, the annual slip is as large as 7 m, corresponding to ~12 pixels if the slip occurs along the azimuth direction and ~4 pixels if along the range direction. With an increasing number of years, the ~21-m slip over a course of 3 years corresponds to ~35 pixels if the motion occurs along the azimuth direction and ~13 pixels if along the range direction. When we apply a matching image patch of $48 \times 48$ pixels, the overlapping area between two image patches is too small for an effective cross-correlation, in particular for the multi-year azimuth offset tracking for the landslide-parallel tracks T03501 and T21501. Therefore, we use intermediate temporal intervals (0.5–2.5 years) that fit the resolvability considering the in situ image resolution and displacement rate, so as to ensure the displacements during the time are smaller than the size of the image patch to avoid aliasing. We apply $48 \times 48$-pixel patches for the fly-along-slip tracks (T03501 and T21501), and $32 \times 32$-pixel patches for the fly-across-slip tracks (T30502 and T12502). For each track, there are about 100+ offset tracking pairs (Supplementary Fig. 1 and Supplementary Table 1), allowing us to suppress artefacts and derive displacement time series.

**Enhanced 3D displacements and time-series analysis.** One unique aspect of the UAVSAR data sets is that four independent tracks image the landslide on the same date. Independent measurements along both azimuth and range directions can be extracted from the constrained-temporal-interval POT method. We have a more than sufficient number of data sets to constrain the 3D displacements with high confidence.

To achieve the 3D time series and velocity, we use the following approach.

First, we set up a grid frame ($50 \times 50$ m for each grid) in the geographic coordinate system. We reference the data to the same stable area to standardize the motion magnitude and resample the data to the grid to standardize the positions.

Second, we calculate the time-series displacements for eight measurable directions (two for each track; four tracks). We only consider the grid elements with the number of valid measurements exceeding 90% of the total available number. For each grid element from each measurable direction, we have:

$$G_{m,n}^{\text{radar}} \times t_{n,1}^{\text{radar}} = d_{m,1}^{\text{radar}}, \tag{2}$$

where $m$ is the number of data pairs, $n$ is the number of acquisition dates, $G_{m,n}^{\text{radar}}$ is a sparse matrix that indicates the sequence of master and slave dates, $d_{m,1}^{\text{radar}}$ is the SAR-derived displacement measurements, and $t_{n,1}^{\text{radar}}$ represents the time-series displacements to be solved.

An enhanced time-series displacement along the range direction is achieved from combining the short-time-interval InSAR measurements and relatively long-time-interval pixel offsets while the azimuth displacements are only from the POT method.

Third, we calculate the 3D time series from the derived azimuth and range time series $t_{n,i}^{radar}$, where $t_{n,i}^{radar} = [t_{n,1}^{radar}, t_{n,2}^{radar}, ..., t_{n,i}^{radar}]$ ($3 \leq i \leq 8$). For each grid element, we have:

$$t_{n,3}^{3d} \times G_{3,i}^{3d} = t_{n,i}^{radar}, \tag{3}$$

where $G_{3,i}^{3d}$ represents the radar unit vectors, and $t_{n,3}^{3d}$ is the unknown 3D displacements. The velocity can be estimated from the linear fit of the time series.

**Exponential time-series model**. We characterize the multi-annual behavior by representing the longer-term displacement rate changes as an exponential function of time:

$$d(t) = M(e^{kt} - 1), \tag{4}$$

where $d(t)$ is the displacement at time $t$ starting from 0, the exponent $k$ can describe both accelerating ($k > 0$) and decelerating ($k < 0$) phases, $M$ is the magnitude coefficient and its sign is determined by the overall displacements. For an accelerating behavior ($k > 0$), $M > 0$ is for an increasing trend, and $M < 0$ for a decreasing trend, and vice versa. The exponent $k$ highlights the rate changes, and here we define it as the velocity change with respect to its initial values. The exponential model allows us to first identify the multi-annual acceleration and deceleration phases and then quantify the rate changes. Fig. 5 and Supplementary Figs. 16 and 17 show the model results with the input of net 3D motions $(E^2 + N^2 + U^2)^{\frac{1}{2}}$.

**Three-dimensional viscoplastic flow**. We simulate the earth flow movement in an inclined channel (Supplementary Fig. 7). Rheologically, the Slumgullion landslide consists of viscoplastic materials. Assuming a Herschel–Bulkley model rheology[35], the bottom layer behaves like a fluid when the shear stress exceeds the yield stress, and the top layer translates like a rigid plug where the shear rate is nearly 0. This constitutive law is described by:

$$\tau = \tau_c + K\dot{\gamma}^n, \tag{5}$$

where $\tau_c$ is the yield stress at the yield interface $z = h_0$ (Supplementary Fig. 7b), $\dot{\gamma}$ is the shear strain rate, $K$ is the consistency index with dimensions of Pa s$^n$, and $n$ is a dimensionless flow index that characterizes how non-Newtonian the flow is. For $n < 1$, the flow is shear thinning and viscosity decreases with increasing shear strain rate and stress[36]. This law generalizes the Coulomb material with $K = 0$ and $\tau_c = \sigma$ʹ$\tan \varphi$ where $\varphi$ is the bulk friction angle; the Bingham fluid with $n = 1$ and constant $\tau_c$; and the power-law fluid with $\tau_c = 0$. We choose the power-law theory in this study and we give the reasons below.

The governing equations are given by the mass and momentum balance equations, supplemented by a no-slip condition at the basal bed, i.e., downslope velocity $u = 0$ at $z = H$, and no shear stress at the free surface, i.e., $\tau = 0$ at $z = D$.

Assume that, first, the transverse velocity vanishes everywhere, $u_y \equiv 0$; and second, the mass flow occurs in a layer of constant thickness for the infinitesimal distance downhill, and thus $\frac{\partial h}{\partial x} = 0$. For the power-law flow, the longitudinal velocity profile can be simplified as[23]:

$$u_x = \frac{n}{n+1} \left( \frac{\rho g \sin \alpha}{K} \right)^{\frac{1}{n}} \left[ (D-H)^{\frac{n+1}{n}} - (D-z)^{\frac{n+1}{n}} \right] \text{ for } H \leq z \leq D. \tag{6}$$

**Channel configuration**. We propose a cross-sectional form of the landslide (Supplementary Figs. 8b and 9) that captures the situation with a flattening bed, which is highly likely the case for many sections in the Slumgullion landslide based on the shape of the velocity profiles and field evidence[11]:

$$H = D_0 - \frac{\tan^{-1}\left(\left(y + \frac{L}{2}\right) \times s\right) + \tan^{-1}\left(\left(-y + \frac{L}{2}\right) \times s\right)}{2\tan^{-1}\left(\frac{L \times s}{2}\right)} \left(1 + \frac{2y \times \sin\beta}{L}\right) D_0, \tag{7}$$

where $D_0$ is the depth at the central axis, $s$ describes the steepness from the basal beds to the lateral flanks, and $\beta$ allows the bed to tilt and thus the channel to be asymmetric. $L$ represents the landslide width and can be readily extracted from the displacement map. We invert the channel geometric parameters from the viscoplastic flow model using the least-squares estimates. We utilized 57 out of 77 transverse velocity profiles, with 20 other profiles being excluded that are in the upper head zone where the SNR is too low (Fig. 2 and Supplementary Fig. 4). We can reasonably assume a listric failure surface[14] with effectively 0 depth, steepness, and tilt at the uppermost head.

**Choice of power-law flow model and parameterization**. A borehole inclinometer[37] (located 17 m into the landslide normal to the bounding fault from the middle extensometer monitoring site) reveals negligible motion from the ground surface to 5-m depth measured between October 19, 2016 and December 7, 2016, and only ~20-mm of shear deformation occurred between about 5- and 10-m depth while the ground surface moved by ~0.6 m with respect to outside of the landslide

from September 4, 2016 to October 17, 2016 (Supplementary Fig. 6). This provides strong evidence for the existence of an effectively undeforming plastic plug in the upper part of the debris slide. On the other hand, SAR data and extensometers[37] have confirmed shearing at the shallow lateral flanks (Figs. 2 and 3). The classic Bingham flow model, with the presence of a yield surface at which flow terminates, may not be suitable for analyzing the entire landslide. Nonetheless, the Bingham flow model can be used to quantify the intrinsic viscosity of the frontal toe area[38]. Here, we consider a power-law flow regime to allow for the formation of a pseudo-plug at shallow depth and the development of lateral shear zones.

As the slide is mostly saturated, we consider the density for the saturated condition[2] $\rho = 1468 \text{ kg m}^{-3}$ and an average slope $\alpha = 8°$ in Eq. (6). The constitutive rheology parameters can be inferred if we know the slide depth at the known velocity profiles. The only reliable depth estimates that exist are at the toe. Because the emergent toe moves over the old, undeforming ground surface, we can simply compare the topography on and off the slide toe (Supplementary Fig. 5; ref. [39]). The depths of transverse profiles #76 and #77 are estimated to be 17.2 and 12.8 m, respectively. Then we constrain the constitutive parameters $K$ and $n$.

Given the strong trade-offs between the flow index $n$ and the consistency index $K$, we constrain the best-fit $K$ at given $n$ (e.g., 0.4–1) between the modeled and observed velocity profiles (Supplementary Table 2). Large $n$ (>0.8) fails to produce the velocity profiles of #76 and #77 at the anticipated depth, which is reasonable because when $n$ is close to 1, the flow behaves more like a Newtonian flow, in contrast with the inferred rheology and observations at Slumgullion. On the other hand, small $n$ (<0.6) produces too high velocity even at shallow depth (Supplementary Fig. 11). Therefore, we apply the flow index $n$ as 0.7, and the best-fit consistency index $K$ as $1.34 \times 10^{10} \text{ Pa s}^n$ for the whole landslide.

**Fluid water recharge**. Snowmelt and rainwater are the major sources of fluid water recharge to the system. To obtain the fluid water recharge, we need to separate out the rainfall and snowfall from the precipitation, and also extract the snowmelt from the snow water equivalent (SWE) at the US National Resources Conservation Service Slumgullion SNOTEL site. We apply the commonly used linear transition model to estimate the snow fraction in the precipitation based on the air temperature[40]. The daily snow $S$ (mm) is given by:

$$S = \begin{cases} 0 & \text{for} \quad T_{air} \geq T_{rain} \\ P_t \left( \frac{T_{rain} - T_{air}}{T_{rain} - T_{snow}} \right) & \text{for} \quad T_{snow} < T_{air} < T_{rain}, \\ P_t & \text{for} \quad T_{air} \leq T_{snow} \end{cases} \tag{8}$$

where $P_t$ is the daily precipitation total, $T_{air}$ is the average daily temperature (°C), $T_{rain}$ and $T_{snow}$ are the air temperatures thresholds (°C); above $T_{rain}$ all precipitation falls as rain and below $T_{snow}$ as snow. We apply 4 °C and 0 °C for $T_{rain}$ and $T_{snow}$, respectively, to best fit the occurrence of abundant SWE decrease at the Slumgullion.

We further infer the snowmelt from the SWE data. We consider the daily sampled SWE as the sum of the remaining snow from the previous days and the fresh snow received on the date, from which that of melt is subtracted. Therefore, the daily snowmelt can be represented as $M = S - \text{diff}(SWE)$. We do not allow for negative snowmelt values likely due to artefacts from wind[41]. The fluid water recharge is the sum of $P_t$ and $M$ (Supplementary Fig. 12).

**Pore pressure diffusion model**. Fluid water recharge modulates the subsurface pore pressure, Coulomb frictional strength, and landslide rates. Transient pore pressure changes can be characterized by the one-dimensional diffusion model[27,42]:

$$\frac{dP}{dt} = D \frac{d^2P}{dz^2} \quad t > 0, z > 0, \tag{9}$$

where $P$ is the transient pore pressure, $t$ is time, $D$ is the effective hydraulic diffusivity, and $z$ is depth below the ground surface. This model describes the downward propagation of pore pressure waves and captures the first-order behavior of pore pressure changes.

The transient pore pressure near the surface is in essence modulated by fluid water recharge $R(t)$ on the surface ($z = 0$):

$$P = r \times R(t) \text{ at } z = 0. \tag{10}$$

An empirically calibrated infiltration scaling factor $r$ scales the water to a pressure value. An analytical solution is given by[42,43]:

$$P(z,t) = \frac{z}{2\sqrt{\pi D}} \int_0^t \frac{e^{-\frac{z^2}{4D(t-s)}}}{\sqrt{(t-s)^3}} P(s, z=0) ds. \tag{11}$$

Groundwater at Slumgullion typically has been observed to be ~2 m below the ground surface near the middle part of the landslide[10,12]. This yields a pore pressure of 177 kPa at the assumed landslide basal depth of 20 m. The hydraulic diffusivity is known to be $7.8 \times 10^{-5} \text{ m}^2 \text{ s}^{-1}$ (ref. [2]). We solve for $r$ that fits the median pore pressure of 177 kPa at 20-m depth and apply it to estimate the pore pressure variations for the other depths (Supplementary Fig. 13).

## Data availability

UAVSAR SLC stacks are available from NASA/JPL (https://uavsar.jpl.nasa.gov). Sentinel-1 data are archived at the Copernicus Open Access Hub (https://scihub.copernicus.eu) and Alaska Satellite Facility (https://search.asf.alaska.edu). The Slumgullion LiDAR DEM (https://doi.org/10.5069/G91834KD) is distributed by OpenTopography (https://opentopography.org). LiDAR data acquisition and processing completed by the NSF funded National Center for Airborne Laser Mapping. Meteorological data are available from US Natural Resources Conservation Service (https://wcc.sc.egov.usda.gov/nwcc/site?sitenum=762). The extensometer and inclinometer data sets are included in the supplement.

## Code availability

The data analysis and processing were conducted with the commercial software MATLAB. The various scripts for data analysis are available from the authors upon request.

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

## Acknowledgements

We thank Noah J. Finnegan, Alexander L. Handwerger, and Matthew A. Thomas for their thoughtful suggestions prior to submission. We thank the UAVSAR flight and data processing teams for acquiring the data and processing the SLC stacks. We thank all the data providers—UAVSAR, Sentinel-1, LiDAR DEM, meteorological data can be freely downloaded from the NASA/JPL, Copernicus Open Access Hub and Alaska Satellite Facility, OpenTopography, and US Natural Resources Conservation Service, respectively. This work contains modified Copernicus data from the Sentinel-1A and -1B satellites processed by the European Space Agency (ESA). The figures were produced using MATLAB, ArcGIS, Adobe Illustrator, and the open source program Generic Mapping Tools (GMT). This research was sponsored by the NASA Earth Surface and Interior Focus Area. Part of this research was performed at the Jet Propulsion Laboratory, California Institute of Technology under contract with NASA. Any use of trade, firm, or product names is for descriptive purposes only and does not imply endorsement by the US Government.

## Author contributions

X.H. and R.B. designed the study. X.H. analyzed SAR images and performed the modeling. W.H.S. collected and interpreted the field data. E.J.F. planned and coordinated UAVSAR data collection and analysis. X.H. drafted the paper with input from all co-authors.

## Competing interests

The authors declare no competing interests.
