## [Peer Review File · Nature Communications]

Reviewers' comments:

Reviewer #1 (Remarks to the Author):

Review of "Surface and subsurface dynamics of a perennial slow-moving landslide from ground, air and space » by Xie Hu, Roland Bürgmann, William H. Schulz, and Eric J. Fielding

This paper studies the motion of the Slumgullion landslide in Colorado using hybrid remote sensing data (Sentinel 1 and UAVSAR) and three extensometers located in the landslide area. The authors are able to extract the velocity field of the slide between 2011 and 2018. Using a two layers mechanical model, they are able to infer the geometry and rheology of the landslide.

I found the paper very interesting, well written with clear figures. I am unable to judge the SAR measurements but rely on the high expertise of the authors. I will just comment on the mechanical model. I find the paper suitable for publication in Nature Geosciences once the authors have answered to the comments mentioned below, that I think are minor.

The mechanical model is described in the supplements (Fig S7). I think that this simple model captures the essential features of the dynamics of the landslide: a rigid plug moves on top of a viscous layer with non Newtonian rheology. The channel model (Fig S8a) is converted into a geometry (Fig S8b) more consistent with the actual geometry of the Slumgullion landslide. I have the following questions concerning the modeling:

1) Eq S5: Eq. (S5) contains the yield stress τ_c . If this yield stress is a frictional stress, then τ_c should read $\tau_c = \mu\sigma$, where μ is the friction coefficient and $\sigma = \sigma_0 - p$ should be the effective normal stress, σ_0 being the normal stress due to the overburden material and p the pore pressure. It is not clear to me if the yield stress is used in the modeling. If yes, what are the values of μ and is the pore pressure taken into account? If not, could you explain why it is not used?

2) Channel configuration: my feeling is that formula S7 is incorrect. The second term on the right hand side of the equation is dimensionless while it should have the dimension of a length like D_0 . I would expect the function $H(z)$ to verify $H(-L/2) = H(L/2) = 0$ and $H(0) = D_0$ which is not the case. Please correct this.

3) Channel configuration parameters: what are the parameters obtained by least square adjustments? D_0 , L , s and β ? I found no table showing the values of the obtained parameters and no discussion on the reliability of those estimates. Please provide this.

4) Pore pressure diffusion model: this part of the modeling is crucial and I had troubles understanding the model in detail. The 1D diffusion model is simple and I have no questions on this part. For the pore pressure part, I understand that the fluid water recharge is converted to a pore pressure distribution at the surface $P(z=0)$ using an infiltration factor r (S10). Giving (S10) as initial condition to (S9) could allow the derivation of the time evolution of pore pressure with depth, if the bottom depth of the model is known (which should be the case from the Channel configuration part). I don't understand how the hydraulic diffusivity and the median pore pressure are obtained based on the observation that ground water lies 2m from the ground surface. Please detail this part.

Hugo Perfettini

Reviewer #3 (Remarks to the Author):

The main claim of this article is a rather deep investigation of a single landslide merging the experimental data obtained by 2 satellite passages (SAR operating in C-band), 4 flights with an airborne L-band SAR and 3 extensometers. All the used remote sensing techniques are well-known. Many landslides in the world are monitored in the same way. I have to precise that I'm an expert of remote sensing, not an expert of fluid dynamics, so I am not able to judge the model the authors use for analyzing the dynamic behavior of the landslide. Nevertheless, correlations between hydrometeorological data and displacement are well known in literature. Therefore, as far as my field of expertise is concerned (remote sensing of landslides), I do not see a major step ahead in this article. I think that this article could be published more properly in a specialistic journal.

Re: NCOMMS-19-38706A

Reviewer 1: Dr. Hugo Perfettini

This paper studies the motion of the Slumgullion landslide in Colorado using hybrid remote sensing data (Sentinel 1 and UAVSAR) and three extensometers located in the landslide area. The authors are able to extract the velocity field of the slide between 2011 and 2018. Using a two layers mechanical model, they are able to infer the geometry and rheology of the landslide.

I found the paper very interesting, well written with clear figures. I am unable to judge the SAR measurements but rely on the high expertise of the authors. I will just comment on the mechanical model. I find the paper suitable for publication in Nature Geosciences once the authors have answered to the comments mentioned below, that I think are minor.

We appreciate the valuable comments that help us improve our paper. Please find below our detailed responses in bold. We modified the manuscript as documented in a tracked copy we are also including in our resubmission.

The mechanical model is described in the supplements (Fig S7). I think that this simple model captures the essential features of the dynamics of the landslide: a rigid plug moves on top of a viscous layer with non Newtonian rheology. The channel model (Fig S8a) is converted into a geometry (Fig S8b) more consistent with the actual geometry of the Slumgullion landslide. I have the following questions concerning the modeling:

1) Eq S5: Eq. (S5) contains the yield stress τ_c . If this yield stress is a frictional stress, then τ_c should read $\tau_c = \mu\sigma$, where μ is the friction coefficient and $\sigma = \sigma_0 - p$ should be the effective normal stress, σ_0 being the normal stress due to the overburden material and p the pore pressure. It is not clear to me if the yield stress is used in

the modeling. If yes, what are the values of μ and is the pore pressure taken into account? If not, could you explain why it is not used?

Equation S5 generalizes the stress condition for different rheological conditions. To characterize the non-Newtonian behavior, the most commonly used Bingham plastic model relies on the yield stress τ_c and K (viscosity) to govern the stress. This model predicts no-flow zones within a certain distance from the lateral margins (Fig. S8a), where the depth to the surface is too shallow to drive the flow as long as the bed shear stress is below the yield stress (Mei and Yuhi, 2001). However, the extensometers and SAR measurements both confirm appreciable motion at the lateral flanks (Figs. 2 and 3). Therefore, to characterize the non-Newtonian behavior for the entire landslide, we chose the power-law theory for which the yield stress τ_c equals to 0. Therefore, there is no discrete yield surface and the pseudo plug gradually transitions into the yield zone underneath.

There is a section on the “Choice of power-law flow model and parameterization” in the Methods (starting from line 164 in the tracked copy). We added a sentence after Equation S5 (line 134) to refer to this section. The relevant descriptions are in lines 116-121 in the Main text.

Reference:

Mei, C. C., & Yuhi, M. (2001). Slow flow of a Bingham fluid in a shallow channel of finite width. *J. Fluid Mech.*, 431, 135–159.

2) Channel configuration: my feeling is that formula S7 is incorrect. The second term on the right hand side of the equation is dimensionless while it should have the dimension of a length like D_0 . I would expect the function $H(z)$ to verify $H(-L/2)=H(L/2)=0$ and $H(0)=D_0$ which is not the case. Please correct this.

Thank you for your careful reading of the equation. We realized this mistake after submitting the manuscript. The formula associated with trigonometric functions was altered when the document was being uploaded to Google Docs for online editing by the coauthors. Equation S7 should be

$$H = D_0 - \frac{\tan^{-1}\left(\left(y + \frac{L}{2}\right) \cdot s\right) + \tan^{-1}\left(\left(-y + \frac{L}{2}\right) \cdot s\right)}{2 \tan^{-1}\left(\frac{L \cdot s}{2}\right)} \left(1 + \frac{2y \cdot \sin \beta}{L}\right) D_0 \quad (S7)$$

We checked all the equations throughout the text.

3) Channel configuration parameters: what are the parameters obtained by least square adjustments? D_0 , L , s and β ? I found no table showing the values of the obtained parameters and no discussion on the reliability of those estimates. Please provide this.

For the last two transverse profiles #76 and #77, where we can estimate the thickness from the projection of the surface topography from the downslope stable terrane, we used D_0 and the longitudinal velocity profiles u_x as the input to invert for the other geometric parameters L , s , β and the rheological parameters consisting of the flow index n and consistency index K . Table S2 gives the best solution of the rheological parameters (pairs of n and K) and their corresponding residuals between the modeled and observed longitudinal velocities; this helps us rule out the large- n scenarios as the residuals are too large. Fig. S12 shows the longitudinal velocity profiles from the basal bed to the ground surface; this information helps us rule out the small- n scenarios, as the consequent surface velocity is too large over the shallow beds (Methods lines 182-192).

After that, we used the longitudinal velocity profiles u_x and the resolved rheological parameters as the input to invert for all the geometric parameters D_0 , L , s , β for each of the other transverse profiles. Fig. 2e shows the results of

the resolved geometric parameters and the residuals of the longitudinal velocity with symbols and fitting lines in corresponding colors. We added a new table (current Table S3) listing their values.

The above-mentioned inverse problems are solved using non-linear least squares.

We reorganized the section “Inferred landslide channels and subsurface flow” (starting from line 113) in the Main text and added more information for clarification.

4) Pore pressure diffusion model: this part of the modeling is crucial and I had troubles understanding the model in detail. The 1D diffusion model is simple and I have no questions on this part. For the pore pressure part, I understand that the fluid water recharge is converted to a pore pressure distribution at the surface $P(z=0)$ using an infiltration factor r (S10). Giving (S10) as initial condition to (S9) could allow the derivation of the time evolution of pore pressure with depth, if the bottom depth of the model is known (which should be the case from the Channel configuration part). I don't understand how the hydraulic diffusivity and the median pore pressure are obtained based on the observation that ground water lies 2m from the ground surface. Please detail this part.

Sorry about this confusion. The hydraulic diffusivity of $7.8 \times 10^{-5} \text{ m}^2/\text{s}$ is a previously estimated value from compiling records of atmospheric pressure and pore pressure within the slide (supplementary discussion in Schulz et al., 2009). The median pore pressure is based on the water level. As mentioned in the supplement text (the last paragraph), groundwater at Slumgullion typically has been observed to be within ~2 m of the ground surface in the middle part of the slide. For an approximate depth of 20 m to the basal bed, the pore pressure is $1000 \times 9.81 \times (20-2) = 177 \text{ kPa}$. We then solve for the infiltration factor r that can yield a median pore pressure of 177 kPa at 20-m depth and then apply it to

estimate the pore pressure variations at other depths. We modified the text in Main lines 205-206 and Methods lines 223-227 for clarification.

Reference:

Schulz, W. H., Kean, J. W. & Wang, G. Landslide movement in Southwest Colorado triggered by atmospheric tides. Nat. Geosci. 2, 863–866 (2009).

Hugo Perfettini

Reviewer #3 (Remarks to the Author):

The main claim of this article is a rather deep investigation of a single landslide merging the experimental data obtained by 2 satellite passages (SAR operating in C-band), 4 flights with an airborne L-band SAR and 3 extensometers. All the used remote sensing techniques are well-known. Many landslides in the world are monitored in the same way. I have to precise that I'm an expert of remote sensing, not an expert of fluid dynamics, so I am not able to judge the model the authors use for analyzing the dynamic behavior of the landslide. Nevertheless, correlations between hydrometeorological data and displacement are well known in literature. Therefore, as far as my field of expertise is concerned (remote sensing of landslides), I do not see a major step ahead in this article. I think that this article could be published more properly in a specialistic journal.

Thank you for considering the remote sensing and hydroclimatic correlation aspects of our manuscript. The other reviewer helped by carefully checking the modeling part.

As the reviewer mentioned, this paper is not only about data integration of multiple sources of geodetic measurements and precipitation records, presenting a hybrid InSAR and pixel-offset tracking approach, but also the

mechanical investigation and physical modeling. The Slumgullion landslide, notable for its perceptible daily motions for centuries, represents an important natural laboratory to study the dynamics of slow-moving landslides. In terms of the data processing, we illustrated the limitations of InSAR and pixel offset methods, the problems with precipitation data including both snow and rain, and provided our strategies that can be used as a guidance in other studies in general. For the displacement analysis, we applied an exponential function to effectively differentiate the acceleration and deceleration processes and to quantify the sensitivities. We also demonstrated the correlation between the water and landslide motions using the diffusion model. Most importantly, we linked the knowledge gained from the remotely sensed ground displacements to insights about the shallow landslide dynamics based on fluid mechanics theory. Our interdisciplinary study can contribute to gaining new knowledge of other natural and anthropogenic environments and processes, such as volcanic lahars, submarine slides, and industrial slurries, for a better understanding of their potential hazards and impact on the landscape.

We also hope this paper can bring broader attention to landslides, which claim thousands of lives every year and constantly modify the landscape on a global scale. In the Main text, we describe the spatiotemporal behaviors, the response to the precipitation changes, the inferred subsurface geometry, and the general implications for persistent slow-moving landslides in a hopefully accessible style. The technical details are provided in the Methods section.

REVIEWERS' COMMENTS:

Reviewer #1 (Remarks to the Author):

The authors have answered in a satisfactory way to all my comments. I have no hesitation in recommending this paper for publication as it is, and want to congratulate them for the quality of this work.

Hugo Perfettini